# Four Waves of the COVID-19 Pandemic: Comparison of Clinical and Pregnancy Outcomes

**DOI:** 10.3390/v14122648

**Published:** 2022-11-27

**Authors:** Sladjana Mihajlovic, Dejan Nikolic, Milena Santric-Milicevic, Biljana Milicic, Marija Rovcanin, Andjela Acimovic, Milan Lackovic

**Affiliations:** 1Department of Obstetrics and Gynecology, University Hospital “Dragisa Misovic”, 11000 Belgrade, Serbia; 2Faculty of Medicine, University of Belgrade, 11000 Belgrade, Serbia; 3Department of Physical Medicine and Rehabilitation, University Children’s Hospital, 11000 Belgrade, Serbia; 4Institute of Social Medicine, Faculty of Medicine, University of Belgrade, 11000 Belgrade, Serbia; 5Center-School of Public Health and Health Management, Faculty of Medicine, University of Belgrade, 11000 Belgrade, Serbia; 6Department of Medical Statistics and Informatics, School of Dental Medicine, University of Belgrade, 11000 Belgrade, Serbia; 7Clinic for Gynecology and Obstetrics “Narodni Front”, 11000 Belgrade, Serbia

**Keywords:** pregnancy, COVID-19, variants of concern, pandemic waves, maternal and pregnancy outcomes

## Abstract

During the last two and a half years, clinical manifestations, disease severity, and pregnancy outcomes have differed among pregnant patients with SARS-CoV-2 infection. These changes were preceded by the presence of new variants of SARS-CoV-2, known in the literature as variants of concern. The aim of this study is to describe the differences between maternal clinical characteristics and perinatal outcomes among pregnant women with COVID-19 during four waves of the COVID-19 epidemic in Serbia. This retrospective study included a series of 192 pregnant patients who were hospitalized due to the severity of their clinical status of SARS-CoV-2 infection. During four outbreaks of COVID-19 infection in Serbia, we compared and analyzed three sets of variables, including signs, symptoms, and characteristics of COVID-19 infection, clinical endpoints, and maternal and newborn parameters. During the dominance of the Delta variant, the duration of hospitalization was the longest (10.67 ± 1.42 days), the frequency of stillbirths was the highest (17.4%), as well as the frequency of progression of COVID infection (28.9%) and the requirement for non-invasive oxygen support (37%). The dominance of the Delta variant was associated with the highest number of prescribed antibiotics (2.35 ± 0.28), the most common presence of nosocomial infections (21.7%), and the highest frequency of corticosteroid therapy use (34.8%). The observed differences during the dominance of four variants of concern are potential pathways for risk stratification and the establishment of timely and proper treatments for pregnant patients. Early identification of the Delta variant, and possibly some new variants with similar features in the future, should be a priority and, perhaps, even an opportunity to introduce more accurate and predictive clinical algorithms for pregnant patients.

## 1. Introduction

Since the official outbreak of the COVID-19 pandemic in China in December 2019 [1], pregnant women, as well as all other vulnerable population groups, have lived in a vicious cycle of ever-emerging new waves of the pandemic. Serbian authorities officially reported the first COVID-19 case in March 2020 [2]. The initial shock of the unknown was mixed with mandatory strict and restrictive social measures, culminating in a governmental lockdown announcement [3]. In general, hospital capacities for non-COVID patients have been limited, medical professionals have transferred to newly formed COVID hospitals, and changes in healthcare systems have culminated in limited maternal and prenatal care and resources worldwide [4]. According to the World Health Organization (WHO) data, as of 17 November 2022, there were 2,415,439 confirmed cases of COVID-19 and 17,326 deaths attributed to COVID-19 in Serbia; moreover, there were 6,717,622 doses of vaccines administrated as of 4 September 2022 [5].

In the last two and a half years, different strategies and approaches have been introduced in order to prevent and reduce perinatal morbidity and neonatal mortality related to limited antenatal care (ANC) access, since ANC is recognized as the most cost-effective mechanism of prevention [6,7]. Over time, our patients have embraced different models of ANC, such as scheduled appointments, home visiting, self-quarantine, community clinics, and hybrid models [8]. In the meantime, the clinical manifestations and disease severity of severe acute respiratory syndrome coronavirus 2 (SARS-CoV-2) infections have also changed, shifting from asymptomatic to life-threatening conditions, such as acute respiratory distress syndrome (ARDS), respiratory failure, and death [1].

New variants of SARS-CoV-2 have emerged since its genome is prone to mutations [9], showing us that the virus is continuously evolving and challenging our therapeutic strategies. Several routes of transmission have been speculated, including contact via droplets and airborne transmission [10]. Furthermore, in aerosols, the SARS-CoV-2 half-life median estimate time is approximately 1.1–1.2 h [10]. Even though many variants of the virus have quickly vanished, selected advantages have enabled the Alpha, Beta, Gamma, Delta, and Omicron variants of the virus to surpass other variants globally [11]. In September 2020, Alpha was first detected in the United Kingdom; in May 2020, Beta was detected in South Africa; Gamma was detected in Brazil in November 2020; Delta was detected in October 2020 in India; and Omicron was detected in November 2021 [12].

Increased risks for hospitalization, admission to intensive care units (ICU), and mortality have been reported among patients infected with Alpha, Beta, Gamma, and Delta variants compared to the wild type virus. All four variants have different degrees of risk, especially among vulnerable groups. Therefore, Delta was most commonly associated with the risk of admission to ICU and mortality, while hospitalization was most commonly indicated among patients infected with the Beta variant [13].

Delta became the dominant variant in May 2021 [14]. It is reported to be 60% more contagious than the Alpha variant and it has 23 more mutations compared to the Alpha variant [15].

The discovery of the Omicron variant was announced by the officials of multiple countries at the same time. Omicron has led to trajectory changes of the pandemic once again; even though it appears that Omicron causes less severe clinical presentations among infected patients [16], its potential to infect a larger number of people, leading to long-term consequences and symptoms, has raised concerns [17]. Upper respiratory tract infection symptoms, such as a cough, sore throat, fever, headache, and chills, are most commonly associated with the Omicron variant [18].

The aim of this study was to describe the differences between the clinical characteristics and perinatal outcomes among pregnant women with COVID-19 during four waves of the COVID-19 epidemic in Serbia.

## 2. Materials and Methods

### 2.1. Study Design and Sample

The retrospective observational study included 192 pregnant patients who were hospitalized due to the severity of the clinical status of SARS-CoV-2 infection at the tertiary health care level at the University Hospital “Dr. Dragisa Misovic”. The admission criteria to the hospital were established based on an adapted version of the Modified Early Obstetric Warning Score (MEOWS) [19]. No patients from the study group were vaccinated against COVID. Exclusion criteria were multiple pregnancies and a negative PCR test for SARS-CoV-2.

The institutional Review Board of the University Hospital “Dr. Dragisa Misovic”, Belgrade, Serbia, approved the study protocol in August 2020.

### 2.2. Study Setting

During the time of the COVID-19 epidemic in 2020 and 2021, in Serbia, the University Hospital “Dr. Dragisa Misovic” was transformed into a COVID-19 hospital. As such, it served as the Serbian referral center for severely ill pregnant patients infected with COVID-19.

Based on the admission and discharge dynamics of our patients, there were four outbreaks of COVID-19 infection in 2020 and 2021 in Serbia, which coincided with the dominance of Alpha, Beta, Gamma, and Delta variants of SARS-CoV-2 [12]. The first wave of the pandemic lasted from March to August 2020, the second wave lasted from October to December 2021, the third wave lasted from February to May 2021, and the fourth wave from September to November 2021. The first wave coincided with the dominance of the Beta variant of concern (VOC), the second wave with the dominance of the Alfa VOC, the third wave with the dominance of the Gamma VOC, and the fourth wave with the dominance of the Delta VOC.

### 2.3. Study Variables

There were three sets of variables: signs, symptoms, and characteristics of COVID-19 infection (the first set), clinical endpoints (the second set), and maternal and newborn parameters, as well as obstetrical characteristics (the third set).

### 2.4. Signs, Symptoms, and Characteristics of the COVID-19 Infection

Upon hospitalization, patients were interviewed, and the presence or absence of the following signs and symptoms were collected: red or irritated eyes, sore throat, cough, difficulty breathing or shortness of breath, headache, loss of smell or taste, tiredness or diarrhea, antibiotic use before the beginning of hospitalization, as well as the number of days from symptom onset until hospitalization.

### 2.5. Clinical Endpoints

Clinical endpoints were extracted from each patient’s medical chart; they included radiology imaging findings (X-ray and computerized tomography (CT) scan results), D-dimer values, data regarding requirements for non-invasive mechanical ventilation, number of days of non-invasive oxygen requirements, progression of the COVID-19 infection, the day of hospitalization when the peak of deterioration occurred, the number of prescribed antibiotics during hospitalization, the potential use of corticosteroid therapy (methylprednisolone), antiviral drugs (lopinavir/ritonavir or remdesivir), low-molecular-weight heparin (LMWH), and the presence or absence of nosocomial infections as well as acute respiratory distress syndrome (ARDS), severe inflammatory response syndrome (SIRS), shock, multi-organ failure (MOF), pulmonary embolism, and ultimately maternal mortality.

### 2.6. Maternal, Newborns Parameters, and Obstetrical Characteristics

From the maternal parameters, we collected the gestational age at admission to the hospital (calculated in days), delivery mode, parity, and presence of co-morbidities in pregnancy, including gestational diabetes mellitus, gestational hypertension, preeclampsia and anemia in pregnancy, premature rupture of the membrane (PROM), and abnormal uterine bleeding. Maternal anthropometric parameters, pre-pregnancy weight, and body height were collected from the primary health service reports of the patients and pre-pregnancy body mass index (BMI) ranges were calculated.

Additionally, on the first day of hospitalization, ultrasonography parameters were observed, including the amniotic fluid index (AFI), placental maturity grade, intrauterine growth restriction (IUGR), and large for gestational age (LGA).

Newborn parameters included the infants’ Apgar scores in the first and fifth min of life, the incidence of prematurity, and data regarding fetal antenatal maturation with dexamethasone.

### 2.7. Statistical Analysis

The results are presented as absolute (*n*) numbers and percentages (%), as well as mean values (MV) and standard deviation (SD). Furthermore, a 95% of confidence interval (CI) was calculated for the continuous variables. Comparisons among the tested groups of patients between the different waves were conducted by the Kruskal−Wallis and ANOVA test for continuous variables, and the Chi−square test for categorical variables. Statistical significance was set at *p* < 0.05.

## 3. Results

There were no significant differences in the mean values of age between the COVID-19 pandemic waves in the tested pregnant women (*p* = 0.199). The frequencies of the different values of BMI significantly differed between pandemic waves (*p* < 0.001); in the first three waves, overweight pregnant women were most frequently affected (63.9%, 55.3%, and 48.9%, respectively); in the fourth wave, obese (39.1%) and normal weight pregnant (41.3%) women had similar percentages. Gestational age at admission significantly differed between the four waves (*p* = 0.011), as well as the frequencies of prematurity (*p* = 0.002), presence of gestational diabetes (*p* = 0.010), PROM (*p* < 0.001), pregnancy outcome (*p* = 0.019), amniotic fluid index (0.049), placental maturity degree (*p* = 0.013), and fetal antenatal maturation (0.004) (Table 1). Gestational age was the highest during the first wave (266.64 ± 5.05) and the lowest during the third wave (244.80 ± 10.21). Term infants were most frequently present in the first three waves (81.4%, 78.9%, and 81.4%, respectively), while in the fourth wave, preterm and term infants made up 50% of each. Gestational diabetes was most frequent in the third wave (17%), while PROM was in the second wave (28.9%). The highest frequencies of stillbirth (17.4%) and fetal antenatal maturation (29.5%) were noticed during the fourth wave. The highest placental maturity grade was in the second wave (2.66 ± 0.13), while the lowest was in the fourth wave (2.21 ± 0.12) (Table 1).

The number of days of hospitalization (*p* = 0.007), ICU duration (*p* = 0.014), the number of days on oxygen therapy (*p* < 0.001), the peak of deterioration from the beginning of hospitalization (*p* < 0.001), and the number of prescribed antibiotics (*p* < 0.001) significantly differed between the four different COVID-19 pandemic waves. Performed CT (*p* = 0.019), progression of COVID infection (*p* = 0.043), administration of corticosteroids (*p* < 0.001), antiviral drugs (*p* = 0.045), low-molecular-weight heparin (*p* < 0.001), and nosocomial infection (*p* = 0.028) significantly differed between pandemic waves (Table 2). The highest frequency of antiviral drugs was in the first wave (9.8%), while the use of low-molecular-weight heparin was in the third wave (95.7%). The longest duration in the ICU was in the second wave (0.82 ± 0.52 days). The highest frequency of the performed CT was in the third wave (34%). During the fourth wave, the duration of patient hospitalizations was the longest (10.67 ± 1.42 days), the frequency of X-ray-confirmed pneumonia (57.8%) and the requirement for non-invasive oxygen support (37%) were the highest, as well as the longest duration of non-invasive oxygen therapy (5.68 ± 1.45 days), the most frequent progression of COVID infection (28.9%), the longest peak of deterioration from the beginning of hospitalization (7.52 ± 5.61 days), the highest number of prescribed antibiotics (2.35 ± 0.28), the use of corticosteroids (34.8%), and the presence of nosocomial infections (21.7%).

The number of days from symptom onset to hospitalization (*p* < 0.001), frequencies of loss of smell (*p* < 0.001) and taste (*p* < 0.001), and tiredness (*p* = 0.010) significantly differed between the four COVID-19 pandemic waves (Table 3).

Loss of smell (60.5%) and loss of taste (60.5%) were the most frequent in the second wave, while tiredness was the most frequent in the fourth wave (56.5%). The longest duration from symptom onset to hospitalization was in wave 3 (5.64 ± 0.55) (Table 3).

## 4. Discussion

During the four waves of the COVID-19 epidemic in Serbia in 2020–2021, we hospitalized and treated 192 severely or critically ill pregnant patients infected with SARS-CoV-2.

In the pregnant patients, the frequency of common early symptoms related to the COVID-19 infection, such as red or irritated eyes, sore throat, cough, difficulty breathing, headache, and diarrhea, unlike tiredness and loss of taste and smell, was similar during the four waves of the epidemic. Tiredness was the most frequently observed early symptom among patients in the third and the fourth groups of patients, while loss of smell and taste were the more frequently observed symptoms of infection in the second group of patients, meaning that Delta and Gamma VOCs were associated more commonly with systemic manifestations of infection compared to the Alpha VOC. Globally, the Delta VOC has resulted in substantially higher rates of cases of hospitalization and deaths [12], which raises the necessity for early clinical assessment, specific monitoring, and surveillance of these patients. Based on the results from the United Kingdom, the proportion of pregnant patients who experienced progression of disease severity has significantly increased during the dominance of Alpha and Delta VOCs; 35.8% of all pregnant women experienced severe disease during the dominance of the Alpha VOC compared to 45.0% during the Delta period [20]. Results from the United States confirmed that the Delta VOC was associated with a more common progression of infection as well as with the rising proportions of patients requiring hospitalization [21]. Our results undoubtedly confirm these statements. During the dominance of the Delta VOC (the fourth wave), we observed that the average duration of hospitalization was nearly double compared to the other three waves. We observed a similar pattern of distribution in the likelihood of the progression of COVID-19 infection during the Delta wave. Patients had up to three times longer average durations of ICU and were more likely to depend on various modes of mechanical ventilation. In the first group of patients, during the dominance of the Beta wave, the peak of deterioration occurred (on average) on the third day from the symptom onset, and it overlapped with the beginning of hospitalization. In the remaining three groups, the peak of deterioration occurred on the seventh day, and hospitalizations were, on average, initiated on the fifth day. Additionally, Ong et al. concluded that compared to Delta and Alpha, the Beta VOC was less likely to lead to the development of pneumonia or a severe form of COVID-19 infection [22].

Patients treated during the Beta wave also had less frequently used antibiotics before hospitalization. These findings suggest the need for the analysis of whether antibiotics were effective in preventing or developing a severe form of COVID-19 infection. Furthermore, in this group of patients, nosocomial infections were less present, which is a constant reminder that the rational use of antibiotics is a safe and proven way of preventing hospital-acquired infections [23].

During the four waves of the COVID-19 epidemic, several different therapeutic strategies were applied in Serbia and the rest of the world [24]. Corticosteroid therapy was initiated in cases of severe maternal infection. Every third patient during the Delta wave was treated with corticosteroid therapy. The recent meta-analysis has proven the effectiveness of corticosteroid therapy among critically ill patients [25], and among the majority of authors, methylprednisolone was designated as the therapy of choice for pregnant women [26].

The initiation of any type of antiviral drug therapy during pregnancy remains controversial [27]. In our hospital, we initiated this type of therapy in selected cases; the decisions for the initiations were made by perinatologists in coordination with intensive care medicine specialists, pulmonologists, and infectologists. Antiviral drugs were prescribed during the first and fourth waves of the pandemic. Lopinavir and ritonavir are protease inhibitors, and they have low placental transfer [28]. They were administered during the first wave of the pandemic, and remdesivir was administered during the fourth wave of the pandemic. These patients were under constant monitoring. We did not detect any adverse reactions or side effects, but due to the low number of patients, our findings remain inconclusive, as do other authors’ results and systemic reviews [29].

Prior to the Royal College of Obstetrics and Gynecology (RCOG) recommendations, which introduced the widely adopted routine administration of low-molecular-weight heparin (LMWH) therapy [30], we administrated LMWH therapy primarily based on laboratory values of D-dimer. Measurements of D-dimer combined with imaging testing were part of the diagnostic algorithm for the diagnosis of pulmonary embolisms [31]. LMWH is not contraindicated during pregnancy, it has comparable efficiencies compared to unfractionated heparin, which has fewer major bleeding complications [32]. Less than a third of patients during the first wave of the pandemic received LMWH, while during the remaining three waves of the pandemic, it was administrated almost regularly, with the highest incidence of 95.7% during the third wave of the pandemic. Patients infected with SARS-CoV-2 during the third wave of the pandemic had the lowest average values of D-dimer, a finding that initially encouraged us to believe that routine administration of LMWH is an effective therapeutic strategy for the prevention of pulmonary embolism. Unfortunately, this has not proven to be true, since one of the two cases of pulmonary embolism was observed during the dominance of the Gamma VOC in the third wave of the pandemic; once again, the low sensitivities and specificities of D-dimer tests during pregnancy were confirmed [33]. These findings emphasize the importance of clinical algorithms, such as the pregnancy-associated YEARS diagnostic algorithm in the diagnosis of pulmonary embolism [34].

During the dominance of Alpha, Beta, and Gamma VOCs, based on pre-pregnancy BMI values, the majority of patients in this study group were overweight. However, results from our study showed that from the first to the fourth wave there was a decrease in the percentage of overweight pregnant patients and an increase in the percentage of obese pregnant patients. During the Delta wave, the obese and normal-weight groups of patients had similar distributions. The pre-pregnancy BMI weight range classified nearly 40% of these patients as obese. Such findings might lead to the possible assumption that in the different waves of COVID-19 infections, pregnant women could have different susceptibilities with regard to their pre-pregnancy BMIs. Obesity is a well-known risk factor for the progression and severity of COVID-19 infection [35,36]. The high incidence of obese patients during the Delta wave might be an explanation for the often worse clinical outcomes among patients during this wave of the epidemic, since certain metabolic phenotypes, as well as obesity, raise susceptibility to poor COVID-19 outcomes [37]. Obesity is a risk factor commonly associated with co-morbidities in pregnancy [38]. Even though the incidence of gestational diabetes differed between the compared groups, it was more frequent during the dominance of the Gamma VOC; even though the incidence of gestational hypertension and preeclampsia did not differ between the four compared groups, during the Delta wave, the likelihood of gestational hypertension was 30% higher. Co-morbidities in pregnancy are by now very well-known risk factors for adverse maternal and pregnancy outcomes among pregnant COVID-19 patients [39,40,41].

The majority of observed patients in this study group were in the late third trimester of pregnancy, a finding which is consistent with other results and confirms that women are at higher risk of COVID-19 infection during the third trimester of pregnancy [42].

Prematurity has increased globally and it was considered to be the leading cause of neonatal morbidity and mortality in the last two decades [43]. Based on recent official reports, the incidence of prematurity in Serbia has reached 12% [44]. Unfortunately, the severity of infection has led to higher rates of prematurity, especially iatrogenic prematurity; these conclusions are supported by other authors’ findings as well [45]. Prematurity affected half of the patients during the Delta wave of the pandemic. The remaining three groups had approximately the same range of premature (20%). This discrepancy in the prematurity rate also explains the difference in the frequency of antenatal corticosteroid maturation therapy between the four compared groups.

Aside from the higher rates of prematurity, during the Delta wave, we observed a devastating incidence of stillbirths that rose to nearly 20%. DeSisto et al. reported a stronger positive connection between the Delta VOC and stillbirth compared to other VOCs [46], which makes a further analysis of the impact of the Delta VOC on fetal well-being essential.

This study has several limitations, including the sample size and the fact that all participants in this study belonged to the Serbian population. The study method is descriptive and, therefore, it is not possible to analyze the cause–effect of patients’ features on the outcomes of pregnancy.

## 5. Conclusions

During the dominance of the Delta VOC, we observed differences in the likelihood of progression of the clinical presentation of COVID-19 infection, raising concerns about the development of a severe form of COVID-19 infection as well as the risk of adverse maternal and pregnancy outcomes. The observed differences during the dominance of the four VOCs (Alpha, Beta, Gamma, and Delta) are potential pathways for risk stratification and the establishment of timely and proper treatments for pregnant patients. Therefore, early identification of the Delta VOC, and possibly some new VOCs with similar features in the future, should be priorities, and perhaps even opportunities for the introduction of more accurate and predictive clinical algorithms for pregnant patients.

## Figures and Tables

**Table 1 viruses-14-02648-t001:** Distribution of maternal and newborn parameters, and obstetrical characteristics regarding the epidemic wave.

		Wave 1 *n* = 61	Wave 2 *n* = 38	Wave 3 *n* = 47	Wave 4 *n* = 46	*p*-Value
**Age, MV ± SD ** (95% CI)		29.62 ± 5.87 (28.12–31.13)	31.47 ± 4.29 (30.07–32.88)	31.60 ± 5.33 (30.03–33.16)	30.52 ± 5.28 (28.95–32.09)	0.199 *
BMI, N (%)	Normal weight	14 (23%)	8 (21%)	12 (25.5%)	19 (41.3%)	<0.001 **
	Overweight	39 (63.9%)	21 (55.3%)	23 (48.9%)	9 (19.6%)	
	Obese	8 (13.1%)	9 (23.7%)	12 (25.5%)	18 (39.1%)	
Parity	1	31 (50.8%)	20 (52.7%)	20 (42.6%)	21 (45.7%)	0.442 **
	2	21 (34.4%)	11 (28.9%)	23 (48.9%)	13 (28.3%)	
	3	8 (13.1%)	6 (15.8%)	3 (6.4%)	11 (23.9%)	
	4	1 (1.6%)	1 (2.6%)	1 (2.1%)	1 (2.1%)	
Gestational age at admission MV ± SD (95% CI)		266.64 ± 5.05 (256.55–276.73)	252.53 ± 10.39 (231.48–273.57)	244.80 ± 10.21 (224.23–265.37)	254.7 ± 4.99 (244.63–264.78)	0.011 ***
Prematurity	Preterm	11 (18.6%)	8 (21.1%)	7 (16.3%)	22 (50%)	0.002 **
	Term	48 (81.4%)	30 (78.9%)	35 (81.4%)	22 (50%)	
	Postterm	0 (0%)	0 (0%)	1 (2.3%)	0 (0%)	
Gestational hypertension	Yes	5 (8.2%)	4 (10.5%)	5 (10.6%)	8 (17.4%)	0.513 **
	No	56 (91.8%)	34 (89.5%)	42 (89.4%)	38 (82.6%)	
Preeclampsia	Yes	1 (1.6%)	2 (5.3%)	3 (6.4%)	5 (10.9%)	0.241 **
	No	60 (98.4%)	36 (94.7%)	44 (93.6%)	41 (89.1%)	
Gestational diabetes	Yes	0 (0%)	3 (7.9%)	8 (17%)	3 (6.5%)	0.010 **
	No	61 (100%)	35 (92.1%)	39 (83%)	43 (93.5%)	
Anemia in pregnancy	Yes	22 (36.1%)	14 (36.8%)	19 (40.4%)	16 (34.8%)	0.949 **
	No	39 (63.9%)	24 (63.2%)	28 (59.6%)	30 (65.2%)	
PROM	Yes	2 (3.3%)	11 (28.9%)	3 (6.5%)	2 (4.5%)	<0.001 **
	No	53 (86.9%)	25 (65.8%)	41 (89.1%)	41 (93.2%)	
Pregnancy outcome, N (%)	Livebirth	58 (96.7%)	37 (97.4%)	44 (93.6%)	38 (82.6%)	0.019 **
	Stillbirth	1 (1.7%)	1 (2.6%)	1 (2.1%)	8 (17.4%)	
	Miscarriage	1 (1.7%)	0 (0%)	2 (4.3%)	0 (0%)	
Abnormal uterine bleeding	Yes	1 (1.6%)	1 (2.6%)	0 (0%)	1 (2.6%)	0.767 **
	No	60 (98.4%)	37 (97.4%)	61 (100%)	37 (97.4%)	
Amniotic fluid index MV ± SD (95% CI)		117.38 ± 4.39 (108.59–126.16)	109 ± 5.35 (98.37–120.06)	116.25 ± 5.96 (104.24–128.26)	126 ± 5.18 (116.44–137.37)	0.049 ***
Placental maturity grading MV ± SD (95% CI)		2.45 ± 0.10 (2.25–2.66)	2.66 ± 0.13 (2.39–2.93)	2.44 ± 0.14 (2.15–2.73)	2.21 ± 0.12 (1.97–2.45)	0.013 ***
Intrauterine growth restriction	Yes	4 (6.6%)	1 (2.6%)	1 (2.2%)	2 (4.7%)	0.681 *
	No	57 (93.4%)	37 (97.4%)	44 (97.8%)	41 (95.3%)	
Large for gestational age	Yes	1 (1.6%)	2 (5.3%)	1 (2.2%)	1 (2.3%)	0.733 *
	No	60 (98.4%)	36 (94.7%)	44 (97.8%)	42 (97.7%)	
Fetal antenatal maturation	Yes	6 (9.8%)	2 (5.3%)	4 (8.9%)	13 (29.5%)	0.004 *
	No	55 (90.2%)	36 (94.7%)	41 (91.1%)	31 (70.5%)	
Delivery mode	Spontaneous	6 (10%)	4 (10.5)	5 (10.9%)	7 (16.3%)	0.200 *
	Stimulated	22 (36.7%)	15 (39.5%)	17 (37%)	14 (32.6%)	
	Induction of labor	9 (15%)	3 (7.9%)	0 (0%)	6 (14%)	
	Elective Cesarean Section	12 (20%)	2 (5.3%)	11 (23.9%)	3 (7%)	
	Emergency Cesarean Section	8 (13.3%)	10 (26.3%)	11 (23.9%)	12 (27.9%)	
	Assisted birth	2 (3.3%)	2 (5.3%)	1 (2.2%)	1 (2.3%)	
Apgar score 1st min MV ± SD (95% CI)		8.25 ± 0.22 (7.8–8.7)	7.63 ± 0.41 (6.8–8.46)	7.53 ± 0.45 (6.62–8.44)	7.02 ± 0.49 (6.04–8.01)	0.262 ***
Apgar score 5th min MV ± SD (95% CI)		9.23 ± 0.25 (8.74–9.72)	8.79 ± 0.43 (7.93–9.65)	8.53 ± 0.49 (7.54–9.52)	7.95 ± 0.54 (6.86–9.05)	0.215 ***

* ANOVA test; ** Pearson Chi-Square test; *** Kruskal–Wallis test.

**Table 2 viruses-14-02648-t002:** Distribution of clinical characteristics in pregnant COVID-19 patients regarding the epidemic wave.

		Wave 1 *n* = 61	Wave 2 *n* = 38	Wave 3 *n* = 47	Wave 4 *n* = 46	*p*-Value
Number of days of hospitalization MV ± SD (95% CI)		5.16 ± 0.36 (4.45–5.88)	6.26 ± 0.80 (4.65–7.88)	6.23 ± 0.55 (5.13–7.34)	10.67 ± 1.42 (7.82–13.53)	0.007 *
Number of days in intensive care unit MV ± SD (95% CI)		0.34 ± 0.23 (0.11–0.80)	0.82–0.52 (0.23–1.86)	0.68 ± 0.35 (0.03–1.39)	3.65 ± 1.27 (1.09–6.22)	0.014 *
X-ray confirmed pneumonia	Yes	10 (16.4%)	8 (21.1%)	19 (40.4%)	26 (57.8%)	<0.001 **
	No	51 (83.6%)	30 (78.9%)	28 (59.6%)	19 (42.2%)	
CT performed	Yes	7 (11.5%)	12 (31.6%)	16 (34%)	15 (32.6%)	0.019 **
	No	54 (88.5%)	26 (68.4%)	31 (66%)	31 (67.4%)	
Non-invasive oxygen therapy requirement	Yes	3 (4.9%)	6 (15.8%)	9 (19.1%)	17 (37%)	<0.001 **
	No	58 (95.1%)	32 (84.2%)	38 (80.9%)	29 (63%)	
Number of days of non-invasive oxygen MV ± SD (95% CI)		0.26 ± 0.201 (0.14–0.66)	1.24 ± 0.609 (0.00–2.47)	1.52 ± 0.663 (0.19–2.86)	5.68 ± 1.45 (2.75–8.61)	<0.001 *
Progression of COVID infection	Yes	5 (8.2%)	7 (18.4%)	7 (14.9%)	13 (28.9%)	0.043 **
	No	56 (91.8%)	31 (81.6%)	40 (85.1%)	32 (71.1%)	
The peak of deterioration from begging of hospitalization (day)		3.16 ± 2.57 (2.51–3.82)	6.78 ± 4.97 (5.13–8.44)	7.51 ± 3.68 (6.43–8.59)	7.52 ± 5.61 (5.86–9.19)	<0.001 *
Number of prescribed antibiotics MV ± SD (95% CI)		1.26 ± 0.114 (1.03–1.49)	1.68 ± 0.239 (1.20–2.17)	1.74 ± 0.179 (1.39–2.10)	2.35 ± 0.28 (1.79–2.91)	<0.001 *
Corticosteroids	Yes	2 (3.3%)	4 (10.5%)	3 (6.4%)	16 (34.8%)	<0.001 **
	No	59 (96.7%)	34 (89.5%)	44 (93.6%)	30 (65.2%)	
Antiviral drugs	Yes	6 (9.8%)	0 (0%)	0 (0%)	3 (6.5%)	0.045 **
	No	55 (90.2%)	38 (100%)	46 (100%)	43 (93.5%)	
Low-molecular-weight heparin	Yes	20 (32.8%)	32 (84.2%)	45 (95.7%)	36 (78.3%)	<0.001 **
	No	41 (67.2%)	6 (15.8%)	2 (4.3%)	10 (21.7%)	
Nosocomial infection	Yes	3 (4.9%)	4 (10.5%)	3 (6.4%)	10 (21.7%)	0.028 **
	No	58 (95.1%)	34 (89.5%)	44 (93.6%)	36 (78.3%)	
ARDS N (%)	Yes	4 (6.6%)	2 (5.3%)	1 (2.1%)	5 (10.9%)	0.375 **
	No	57 (93.4%)	36 (94.7%)	46 (97.9%)	41 (89.1%)	
SIRS N (%)	Yes	1 (1.6%)	2 (5.3%)	2 (4.3%)	2 (4.3%)	0.778 **
	No	60 (98.4%)	36 (94.7%)	45 (95.7%)	44 (95.7%)	
Shock	Yes	1 (1.6%)	2 (5.3%)	1 (2.1%)	2 (4.3%)	0.705 **
	No	60 (98.4%)	36 (94.7%)	46 (97.9%)	44 (95.7%)	
MOF	Yes	1 (1.6%)	2 (5.3%)	1 (2.1%)	2 (4.3%)	0.705 **
	No	60 (98.4%)	36 (94.7%)	46 (97.9%)	44 (95.7%)	
Pulmonary embolism	Yes	0 (0%)	0 (0%)	1 (2.2%)	1 (2.2%)	0.554 **
	No	60 (100%)	34 (100%)	44 (97.8%)	45 (97.8%)	
Lethal outcome	Yes	0 (0%)	1 (2.6%)	2 (4.3%)	4 (8.7%)	0.121 **
	No	61 (100%)	37 (97.4%)	45 (95.7%)	42 (91.3%)	

* Kruskal–Wallis test; ** Pearson Chi-Square test.

**Table 3 viruses-14-02648-t003:** Distribution of symptoms before hospitalization in pregnant COVID-19 patients with regard to pandemic waves.

		Wave 1 *n* = 61	Wave 2 *n* = 38	Wave 3 *n* = 47	Wave 4 *n* = 46	*p*-Value
Number of days from symptom onset to hospitalization MV ± SD (95% CI)		3.07 ± 0.582 (1.90–4.23)	5.16 ± 0.63 (3.88–6.43)	5.64 ± 0.55 (4.52–6.75)	5.63 ± 0.84 (3.94–7.32)	<0.001 *
Antibiotics used before hospitalization N (%)	Yes	17 (27.9 %)	23 (60.5%)	35 (74.5%)	28 (60.9%)	0.847 **
	No	44 (72.1%)	15 (39.5%)	12 (25.5%)	18 (39.1%)	
Red or irritated eyes N (%)	Yes	5 (8.2%)	0 (0%)	0 (0%)	2 (4.3%)	0.076 **
	No	56 (91.8%)	38 (100%)	47 (100%)	44 (95.7%)	
Sore throat N (%)	Yes	13 (21.3%)	7 (18.4%)	6 (12.8%)	14 (30.4%)	0.206 **
	No	48 (78.7%)	31 (81.6%)	41 (87.2%)	32 (69.6%)	
Cough N (%)	Yes	21 (34.4%)	16 (43.2%)	26 (55.3%)	25 (54.3%)	0.096 **
	No	40 (65.6%)	21 (56.8%)	21 (44.7%)	21 (45.7%)	
Difficulty breathing or shortness of breath N (%)	Yes	9 (14.8%)	7 (18.4%)	9 (19.1%)	14 (30.4%)	0.241 **
	No	52 (85.2%)	31 (81.6%)	38 (80.9%)	32 (69.6%)	
Headache N (%)	Yes	6 (9.8%)	10 (26.3%)	6 (12.8%)	6 (13%)	0.138 **
	No	55 (90.2%)	28 (73.7%)	41 (87.2%)	40 (87%)	
Loss of smell N (%)	Yes	10 (16.4%)	23 (60.5%)	9 (19.1%)	15 (32.6%)	<0.001 **
	No	51 (83.6%)	15 (39.5%)	38 (80.9%)	31 (67.4%)	
Loss of taste N (%)	Yes	7 (11.5%)	23 (60.5%)	9 (19.1%)	14 (30.4%)	<0.001 **
	No	54 (88.5%)	15 (39.5%)	38 (80.9%)	32 (69.6%)	
Tiredness N (%)	Yes	18 (29.5%)	16 (42.1%)	27 (57.4%)	26 (56.5%)	0.010 **
	No	43 (70.5%)	22 (57.9)	20 (42/6%)	20 (43.5%)	
Diarrhea N (%)	Yes	1 (1.6%)	2 (5.3%)	1 (2.1%)	2 (4.3%)	0.705 **
	No	60 (98.4%)	36 (94.7%)	46 (97.9%)	44 (95.7%)	

* Kruskal–Wallis test; ** Pearson Chi-Square test.

## Data Availability

All data are presented in this study. Original data are available upon reasonable request from the corresponding author.

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
