# Peer review of "Four Waves of the COVID-19 Pandemic: Comparison of Clinical and Pregnancy Outcomes"

_viruses, 2022, doi:10.3390/v14122648_

Round 1

Reviewer 1 Report

This research is good for publication. It is well designed, findings are presented systemically. However, some of the points need to be addressed before the publication of this article. All the points have been mentioned below:

1.There are few grammatical mistakes in the article, so please check the revised version of the manuscript to avoid such mistakes.

2. Add a few lines about causative agent SARS-CoV-2 & Please provide some information on the SARS-CoV-2 and routes of transmission of SARS-CoV-2 in the introduction of the manuscript. Read & cite the paper mentioned below;

Aerosol transmission of SARS-CoV-2: The unresolved paradox. Trav Med Infec Dis 2020; 37:101869.

2. References are not as per the standard format of the journal. Update accordingly.

Rest is ok. Congratulations to all the authors for the good study.

Author Response

Dear Reviewer,

Thank you for your valuable comments that will improve our manuscript. We are sending point by point answers to raised concerns:  

1.There are few grammatical mistakes in the article, so please check the revised version of the manuscript to avoid such mistakes.

To best of our knowledge we have corrected grammatical errors. 

2. Add a few lines about causative agent SARS-CoV-2 & Please provide some information on the SARS-CoV-2 and routes of transmission of SARS-CoV-2 in the introduction of the manuscript. Read & cite the paper mentioned below;

Aerosol transmission of SARS-CoV-2: The unresolved paradox. Trav Med Infec Dis 2020; 37:101869.

We added in introduction section suggestions concerning SARS-CoV-2 transmission modes and half-life estimates. Additionally we inserted suggested reference in the reference list. 

2. References are not as per the standard format of the journal.

The references are in line with the journal proposition.

Thank you,

Authors

Reviewer 2 Report

This study presented a descriptive analysis of 192 pregnant patients during four different waves of SARS-CoV-2 variants. The reported data has some merit and will benefit the scientific community.

Comments:

  • The first paragraph of the introduction is poetic, not scientific. Either improve or delete.
  • Local statistics from Serbia need to be added to the introduction.
  • Can you add a paragraph on vaccination uptake and coverage among pregnant women in Serbia?
  • Do you have the data on the vaccination status of the 192 women enrolled in your study? This information will help you understand the differences in disease severity.
  • Table 1, BMI data, shows a significance value of 0.001. Can you elaborate on the discussion section on how the variant waves are related to this observation?
  • What is the number of women in each wave?What are your inclusion and exclusion criteria?
  • In the Results section, line 166, you wrote, "While in the fourth wave, normal-weight pregnant women were 166 most commonly affected (41.3%),"  can you explain it in the discussion section?

Author Response

Dear Reviewer,

Thank you for your valuable comments that will improve our manuscript. We are sending point by point answers to raised concerns:  

  • The first paragraph of the introduction is poetic, not scientific. Either improve or delete. 

We deleted the first paragraph. 

  • Local statistics from Serbia need to be added to the introduction.

We included local statistics from Serbia according to official WHO data regarding current pandemic situation and vaccination status. 

  • Can you add a paragraph on vaccination uptake and coverage among pregnant women in Serbia?

We have added vaccination status for entire Serbian population. 

  • Do you have the data on the vaccination status of the 192 women enrolled in your study? This information will help you understand the differences in disease severity.

In our study none of observed subjects have received any vaccine agains COVID-19. We added this in method section.

  • Table 1, BMI data, shows a significance value of 0.001. Can you elaborate on the discussion section on how the variant waves are related to this observation?

We added in discussion section concern regarding BMI differences between different waves. 

  • What is the number of women in each wave?What are your inclusion and exclusion criteria?

In results section we included number of women in each wave. In method section we stated exclusion criteria. 

  • In the Results section, line 166, you wrote, "While in the fourth wave, normal-weight pregnant women were 166 most commonly affected (41.3%),"  can you explain it in the discussion section?

In results section we modified this sentence to adequately resemble results from Table 1 and further explained it in discussion section. 

Thank you,

Authors

Round 2

Reviewer 2 Report

Thank you for addressing my comments.